# Management of an Equine Herpesvirus-1 Outbreak During a Multi-Week Equestrian Event

**DOI:** 10.3390/v17050608

**Published:** 2025-04-24

**Authors:** Nicola Pusterla, Kaila Lawton, Samantha Barnum, Katie Flynn, Steve Hankin, David Runk, Eric Mendonsa, Tara Doherty

**Affiliations:** 1Department of Medicine and Epidemiology, School of Veterinary Medicine, University of California, Davis, CA 95616, USA; kolawton@ucdavis.edu (K.L.); smmapes@ucdavis.edu (S.B.); 2US Equestrian Federation, Lexington, KY 40511, USA; kflynn@usef.org; 3Desert International Horse Park, Thermal, CA 92274, USA; steve@deserthorsepark.com (S.H.); david@deserthorsepark.com (D.R.); 4Fluxergy, Irvine, CA 92618, USA; emendonsa@fluxergy.com; 5West Coast Equine Medicine, Temecula, CA 92590, USA; wcequine5@gmail.com

**Keywords:** equine herpesvirus-1, outbreak, show horses, monitoring, qPCR testing, return to competition

## Abstract

The present study reports on the management of an EHV-1 outbreak at a large, multi-week equestrian event with ongoing showing. Within a 48 h period, 8 horses out of a cohort of 38 horses from the same trainer displayed elevated rectal temperatures ranging from 38.4 to 39.0 °C. Initial testing using a point-of-care PCR assay detected EHV-1 in 2/8 horses, with the results being confirmed at a later time by qPCR. As a precautionary measure and because of the inability to isolate the entire at-risk population, the 38 horses were relocated to an equine facility outside the equestrian event for daily monitoring and weekly EHV-1 qPCR testing of nasal secretions. Overall, 22/38 (58%) horses tested EHV-1 qPCR-positive in nasal secretions over the monitoring period of 28 days, with only one additional horse developing fever. Once all 38 horses tested EHV-1 qPCR-negative twice, 7 days apart, 17 horses returned to the equestrian event to compete for the remaining 2 weeks of the circuit. The present study highlights the importance of isolating and testing horses with fever but also subfebrile horses, as EHV-1 can cause silent infection. The relocation of the exposed horses to an outside facility allowed close monitoring of these horses while reducing the risk of direct and indirect exposure to other show horses. The regular testing for EHV-1 through nasal secretions during the outbreak, coupled with proper biosecurity protocols, allowed the safe return of the show horses to the event. The key elements in reducing the spread of EHV-1 were the routine assessment of rectal temperature, early isolation of horses with elevated rectal temperature and on-site EHV-1 PCR testing.

## 1. Introduction

Equine herpesvirus-1 (EHV-1) is considered a prevalent respiratory pathogen predominantly affecting young horses and causing fever, nasal discharge, mandibular lymphadenopathy and occasional cough [1]. On rare occasions, complications such as abortion and neurological disease can occur and relate to a combination of still poorly understood host, viral and environmental factors [2]. In recent years, various outbreaks of equine herpesvirus myeloencephalopathy (EHM) have led to the costly cancelation of equestrian events and the rapid spread of EHV-1 across counties, states and countries as exposed horses are rushed out of the outbreak facility to return to their home barn [3,4,5]. The lack of daily monitoring and early detection and reporting of clinically diseased horses, coupled with minimal to no biosecurity protocols and the lack of appropriate isolation stabling, are factors responsible for the rapid spread of disease during an EHV-1 outbreak [2,3,5,6]. Various equestrian event organizers and national/international equestrian organizations have actively adopted protocols to mitigate the risk of outbreaks. Such protocols focus on the health of show horses (requirement of certificate of health and vaccination against respiratory viruses), daily monitoring and reporting of fever while at the show grounds and biosecurity measures [7]. Equestrian show venues are further encouraged to have access to isolation facilities to hold febrile horses and to test febrile horses for EHV-1 and other respiratory pathogens if needed. Despite all these measures, the spread of respiratory pathogens is still possible, through unnoticed silent shedding by healthy horses. The frequency of silent shedding varies with the horse population, time of the year and specific pathogen tested, but is in the range of 0–4% for show horses [4,8,9,10]. Despite knowing all these dynamic facts and applying various preventive measures, outbreaks at show venues can still occur. That said, the goal of show organizers is to prepare and respond rapidly to an outbreak situation to allow ongoing showing for non-exposed horses. The present study reports on the management of an EHV-1 outbreak involving 38 show horses at a large, multi-week equestrian event, while horses unrelated to the outbreak continued to compete at the circuit.

## 2. Materials and Methods

This study was performed at a large, multi-week-long sport horse show in Thermal, CA, USA (Desert International Horse Park) during the Desert Circuit, which lasted from 8 January to 16 March 2025. The show venue experienced an EHM outbreak during their 2021/2022 season, leading to the cancelation of four show weeks. In order to reduce the risk of a subsequent outbreak and recognize diseased horses in the early stages, the show organizers instituted the following monitoring and biosecurity protocols for the subsequent show seasons: vaccination requirement against EHV-1, health certificate within 10 days of arrival to the show grounds, twice daily rectal temperature monitoring, on-site rapid EHV-1 PCR testing for febrile horses, and the immediate movement to isolation stabling of horses with fever of unknown origin. Further, emphasis was placed on regular cleaning and disinfecting of vacated stalls, barns, and high traffic areas and in reducing direct contact between horses in and outside their respective stalls. Owners and trainers were encouraged not to share equipment between horses and to use individual buckets to provide water.

Every show season since the 2021/2022 EHM outbreak, the authors have been exploring ways to generate data on environmental contamination in order to establish frequencies of EHV-1 silent shedding in show horses and/or their environment. For the 2025 show season, a voluntary study focused on the swabbing of horse trailers. After arriving and unloading the horses, trainers/owners were asked to participate in the study. Using disposable gloves, the entire length of the trailer wall where all horses were tied during transportation was swabbed using individual sponges (Sponge-Stick with 10 mL neutralizing buffer, 3M, St. Paul, MN, USA). The biocide-free cellulose sponges measure 1.5 × 3 inches and are mounted at one end of a stick and prehydrated with neutralizing buffer diluent for the collection of samples. The sponges were labeled with the date and a number identifying the trailer and kept refrigerated and shipped on ice to the laboratory at the University of California at Davis for sample processing and analysis. Additional information pertaining to the number of horses transported, duration of the transportation, distance of the transportation, and origin of the horses (state/county) was recorded for each trailer sampled.

The population of horses involved in the EHV-1 outbreak was composed of 36 adult Warmblood horses and 2 ponies, 33 males (geldings and stallions) and 5 mares, ranging in age from 5 to 18 years (median 8 years). The vaccination history of the 38 horses consisted in the administration of killed-adjuvanted multivalent EHV-1/EIV vaccine (brand unknown) 17–155 days (median 69 days) prior to arrival at the show. The healthy horses arrived on 5 January 2025 and were placed into two barns (8 horses in barn 11 and 30 horses in barn 21). At the time of arrival, the trailer moving all the horses was enrolled in the voluntary trailer study and one swab was collected. The study population remained healthy without any reported fevers until 26 January 2025 when one febrile (39.0 °C) horse with distal limb edema tested EHV-1 PCR-positive for EHV-1. Once multiple horses developed fever and one additional horse tested PCR-positive for EHV-1, all 38 horses were moved to an empty facility outside the show event. The facility was composed of two multi-stall barns. Horses were stabled indoors based on their EHV-1 shedding status with EHV-1 qPCR-positive horses in one barn and EHV-1 qPCR-negative horses in the other barn. In each barn, horses were separated by solid walls but shared common airspace. Proper biosecurity protocols, including foot bath at the entrance of the barns, dedicated personal protective equipment (booties, coveralls, disposable gloves) and separated stall cleaning equipment were used to reduce indirect transmission amongst the study horses. The horses were monitored closely for elevated rectal temperature and nasal swabs were collected approximately every 7 days in order to document their shedding status. Horses testing EHV-1 qPCR-positive in nasal secretions were treated at the discretion of the owner and attending veterinarian with valacyclovir for 7 days (Camber Pharmaceuticals, Piscataway, NJ, USA) starting at their peak nasal shedding. The treatment consisted in the administration of valacyclovir at 30 mg/kg body weight every eight hours for the first six treatments, followed by 20 mg/kg body weight every twelve hours for the subsequent 5 days. Horses testing qPCR-negative twice, 7 days apart, were allowed to come back to the show event and resume showing.

Nasal secretions were collected at multiple time points from the affected horses and barn mates using 6-inch rayon-tipped swabs (Puritan^®^ Sterile Rayon Tipped Applicators, Guilford, ME, USA). The samples were processed for nucleic acid extraction within 24 h of collection using an automated nucleic acid extraction system (QIAcube HT, Qiagen, Valencia, CA, USA) according to the manufacturer’s recommendations. Thereafter, the purified nucleic acids were tested for the presence of the glycoprotein B (gB), and the polymerase (ORF 30) gene of EHV-1 using previously reported real-time TaqMan PCR assays [11]. Quantitative qPCR results for EHV-1 in nasal secretions were normalized against a housekeeping gene (equine glyceraldehyde-3-phosphate dehydrogenase gene) and expressed as number of gB target genes per million cells as previously reported [11]. Initial nasal swabs from febrile horses were screen using a novel point-of-care (POC) EHV-1 PCR assay, according to the manufacturer’s recommendations (Fluxergy, Irvine, CA, USA).

## 3. Results

Out of the 46 trailer swabs collected on 5 January 2025, only one swab tested qPCR-positive for EHV-1. The swab was taken from a 10-horse trailer that had transported 38 horses in multiple trips from a training barn located 70 miles from the show venue. The detection of the qPCR-positive trailer swab did not change the existing biosecurity and monitoring protocols in place and the information was not shared with the trainer. On 26 January 2025 (21 days from the collection of the trailer swab) one horse (horse 1) from the cohort of 38 horses developed fever (39.0 °C) and distal limb edema and this horse tested EHV-1 PCR-positive using the POC PCR assay. This horse was immediately moved to an isolation barn. Within the next 48 h, an additional seven horses (horse 2–8) developed subfebrile (38.4 to 38.5 °C; 4 horses) to febrile (38.6 °C; 3 horses) rectal temperatures with one of these 7 horses testing PCR-positive for EHV-1 using the POC PCR assay. All initial eight EHV-1 POC PCR results were confirmed via qPCR at a later time and showed 100% agreement. Due to an imminent EHV-1 outbreak and the inability to stable all 38 horses in isolation, it was decided to move all the horses to an outside empty facility to further monitor and isolate these horses.

Throughout the 28-day quarantine period (28 January 2025 to 24 February 2025), only one additional horse (horse 19) developed fever (38.7 °C) on one single day (11 February 2025). All the other horses had rectal temperatures within the normal reference range with peak rectal temperatures ranging from 37.5 to 38.3 °C (median 38.0 °C; Table 1). A total of 22/38 (58%) horses tested EHV-1 qPCR-positive from 26 January 2025 to 21 February 2025, with peak viral loads ranging from 26 to 26,121,278 gB genes/million cells (median peak viral load of 37,799 gB genes/million cells). All the EHV-1 qPCR-positive nasal secretions belonged to the N_752_ genotype. Within the first week of the outbreak (26 January 2025 to 3 February 2025), 13/38 (34%) horses tested qPCR-positive for EHV-1, with only 4 horses showing elevated rectal temperature. During the second and third week of the outbreak (11 February 2025 to 14 February 2025), nine additional horses tested EHV-1 qPCR-positive, with only one horse (19) displaying fever. On the fourth week (2.21.28), two previously EHV-1 qPCR-positive horses still tested positive in nasal secretions. On weeks 5 and 6 of the outbreak, all of the tested horses (*n* = 38 for 27 February 2025; *n* = 3 for 5 March 2025) had EHV-1 qPCR-negative nasal secretions. Seventeen horses (horse 1, 2, 5, 10, 15, 16, 17, 22, 24, 28, 29, 30, 31, 33, 36, 37, and 38) were allowed to return to the competition on 27 February 2025 in order to resume competition. The decision to return to the show circuit was left to the discretion of the trainer. Only horses with at least two nasal swabs, collected 7 days apart, testing EHV-1 qPCR-negative were allowed to compete. At the time of return of selected horses, three isolated horses at the outside facility had only one EHV-1 qPCR-negative nasal swab tested.

At the request of their respective owners, 16 EHV-1 qPCR-positive horses (1, 2, 3, 4, 8, 9, 11, 12, 13, 14, 18, 19, 20, 22, 23 and 32) were treated for 7 days with valacyclovir starting at the first detection of EHV-1 in nasal shedding. With the exception of horse 12, all horses treated with valacyclovir experienced a rapid decline in EHV-1 viral load in nasal secretions. The viral load of horse 12 increased from 40,904 to 2,606,189 EHV-1 gB target genes/million cells while under valacyclovir treatment. After reviewing the dose and frequency of drug administration, it was determined that the horse was refractory to oral drug administration and did not receive the full amount of valacyclovir. The valacyclovir treatment was extended, making sure that the horse would receive the appropriate amount of antiherpetic drugs.

In order to determine if EHV-1 spread from the cohort of 38 horses to adjacent barns at the equestrian venue, the rate of sporadic febrile horses testing EHV-1 qPCR-positive was determined, prior, during and after the outbreak. From 5 January 2025 to 25 January 2025, a total of 0/1 febrile horses tested qPCR-positive for EHV-1. From 26 January 2025 to 24 February 2025, the rate of EHV-1 qPCR-positive febrile horses was 3/12, and during the last 2 weeks of the circuit, 1/3 febrile horses tested qPCR-positive for EHV-1. None of the additional EHV-1 qPCR-positive horses appear to have any epidemiological link to the outbreak horses.

## 4. Discussion

Due to a previous EHM outbreak in 2021/2022 at the present equestrian event, biosecurity measures were heightened with the goal for early detection, immediate isolation of all febrile horses and rapid testing for EHV-1 using a POC PCR assay. These strategies, while time-consuming, rely on the active engagement and communication of all participants. The present study showed that the rapid separation of a cohort of 38 show horses and ponies with evidence of active EHV-1 infection, prevented viral spread and any interruption of the show circuit. Further, the monitoring of these horses via daily clinical assessment and weekly qPCR testing permitted them to resume competition after 28 days of isolation.

An interesting feature of this outbreak was the lack of severe clinical disease, as only nine horses displayed mild elevations of rectal temperature [1]. The level of immunity to EHV-1 often determines the severity of disease, with no to mild clinical disease experienced by horses with preexisting immunity [12]. Preexisting immunity also impacts viral kinetics, with lower peak and shorter duration of shedding seen in horses with mild clinical disease and subclinical shedders [13]. It was not surprising that within the cohort of 38 horses, disease expression was similar (i.e., mild to silent infection), and the authors speculate that this observation relates to a similar demographic, use, husbandry and preventive health protocols. Further, these 38 horses were all exposed to identical environment conditions.

Pre-show requirements, such as health certificate, vaccination against respiratory viruses (EHV-1, EIV), temperature logs 7–10 days prior to arriving at the show, negative EHV-1 qPCR test result if a horse or barn has been involved with in a recent EHV-1 outbreak, all aim at allowing only healthy horses with no evidence of recent EHV-1 infection to participate at an equestrian event. It is well understood that such requirements should be tailored to the size and duration of each event. However, even when all of these requirements are followed, the introduction of silent EHV-1 shedders cannot be prevented and while low, the risk of EHV-1 spread needs to always be accounted for. Various studies looking into the rate of silent shedders have determined that between 0–4% of healthy adult show horses shed selected respiratory pathogens [4,8,9,10]. While sporadic and random testing of nasal secretions from show horses is not practical and sustainable, the testing of the environment, such as stalls, can detect pathogen build up and recognize clusters of positive stalls as a reflection of silent spread [10,14,15]. The testing of trailers on arrival was a new concept for this year’s show event. Trailers should be considered a risk for pathogen transmission knowing that they transport multiple horses over long distances, have poor ventilation, and are seldom cleaned and disinfected [9]. Further, the risk of respiratory disease in transported horses can increase as a consequence of immunosuppression and stress associated primarily with opportunistic bacterial proliferation and viral reactivation [16]. The qPCR-positive EHV-1 testing of the trailer that transported the cohort of 38 horses did not change the biosecurity protocols surrounding these horses but instead increased the monitoring and reporting awareness of febrile horses.

Of interest was the observation that at the onset of the outbreak, only 2/8 febrile and subfebrile horses tested qPCR-positive for EHV-1 using a POC PCR assay. Age and immune status against EHV-1 may determine the time and magnitude of fever and its relationship to viral shedding. While commercial EHV-1 vaccines are considered an aid in disease prevention, they can reduce the incidence of clinical disease associated with EHV-1 infection [2,17]. In a recent EHV-1 experimental study, 2-year-old horses responded within 1 day of inoculation with a high primary fever, respiratory signs and high viral nasal shedding [18]. This was in sharp contrast to older horses, for which fever response and onset of shedding was delayed. Further, older horses exhibited significantly lower rectal temperatures compared to young horses [17]. These findings have clinical relevance when monitoring show horses. First, the monitoring of fever is important to mirror clinical status, however, fever is defined by a threshold (≥ 38.6 °C) and some EHV-1 infected horses may stay in a subfebrile range (38.0 to < 38.6 °C), while still shedding EHV-1. Second, some horses with elevated temperature in the peracute stage may not shed enough virus to be detected by qPCR. Therefore, it is important, from a biosecurity standpoint, to monitor the rectal temperature twice daily of at-risk horses and isolate them even when such horses display subfebrile temperatures. This is especially relevant when a cluster of subfebrile horses is identified. From a diagnostic standpoint, it is also important to retest an initial EHV-1 PCR-negative febrile or subfebrile horse 24 to 48 h later.

Little information is available regarding the outcome of early medical intervention in horses infected with EHV-1. The medical management of EHV-1 outbreaks with anti-herpetic, anti-inflammatory and anti-thrombotic drugs is generally aimed at reducing the risk of complications such as EHM [2,19]. Lowering the viral load in lymphocytes/monocytes during cell-associated viremia using anti-herpetic drugs has been one of the strategies to reduce the risk of EHM development in febrile horses. Unfortunately, blood samples were not collected in the study horses in order to document viremia. The goal to treat horses with valacyclovir was to prevent possible complications and reduce viral shedding, therefore reducing environmental contamination and risk of transmission. While previous work on valacyclovir has shown that replication stops immediately after initiation of treatment for alpha herpesviruses [20], the present data do not allow us to draw any conclusions on the benefit of valacyclovir due to the lack of a matched untreated study population.

The isolation of febrile horses and rapid testing and retesting is essential to determine their shedding status and reduce potential transmission and environmental contamination. Every equestrian even should plan to have appropriate isolation capability, as removing contagious horses during the early shedding period is key to prevent pathogen spread [2]. The isolation facility should be separate from the routine housing barns and allow the containment and care of sick horses. In the present study, due to the large cohort of potentially infected horses, it was decided to move these horses to an empty outside facility. Over a period of 20 days (26 January 2025 to 14 February 2025), a total of 22/38 horses tested EHV-1 qPCR-positive in nasal secretions. Peak EHV-1 shedding of clinically infected horses was similar to the peak shedding of subclinically infected horses, supporting previous work showing little difference in EHV-1 viral load amongst various disease forms in field cases [11,21]. The weekly testing of EHV-1 in nasal secretions by qPCR allowed to establish the shedding status of all horses and also allowed the horses to resume competition once they all tested negative twice, 7 days apart. The detection of new EHV-1 qPCR-positive horses up to 18 days from the onset of the outbreak (26 January 2025), highlights the ongoing transmission in this cohort of horses, despite the physical separation of EHV-1 shedders, the treatment of selected horses with valacyclovir and biosecurity protocols. This observation also points to the difficulty to contain an EHV-1 outbreak outside a strict isolation facility and the benefit of PCR testing of all exposed horses at the onset of the outbreak. While the weekly testing of all cohort horses was beneficial in the management of these horses, it did not reduce the isolation period. The recommended quarantine time of 28 days from resolution of clinical signs from the last infected horse appears to be well justified when dealing with an EHV-1 outbreak situation [2]. The authors believe that moving the cohort of exposed horses outside the show venue was key at mitigating the spread of EHV-1 to unrelated horses and allowing the circuit to go on. This is also supported by the overall low rate of febrile horses prior, during and after the outbreak and the detection rate of epidemiologically unrelated EHV-1 PCR-positive febrile horses. Having the capability to test for EHV-1 onsite through the use of a POC PCR assay was crucial in making timely decisions in outbreak management. Also, the testing of all febrile horses for EHV-1 showed that sporadic EHV-1 infections amongst febrile show horses is not uncommon. Such horses need to be isolated and their respective barn mates monitored closely.

As this was a convenience field study, there were various limitations pertaining to the study design, including the overall low number of horses involved in the EHV-1 outbreak, variability in sample collection time points and the lack of a control population. It would have been of interest to test blood for EHV-1 to determine the frequency of viremia within the study population. Further, the measurement of anti-EHV-1 antibodies in blood and nasal secretions would have been of interest to assess the potential protective status of the study horses.

## 5. Conclusions

The present study highlights the importance of immediately isolating and testing horses with fever, but also subfebrile horses, as EHV-1 can cause silent infection. While individual horses can be rapidly moved and managed in an isolated space at the show grounds, the situation is different when an entire barn is at risk of infection. The relocation of the exposed horses to an outside facility allowed for the close monitoring of these horses while reducing the risk of direct and indirect exposure to unrelated show horses. The regular monitoring of EHV-1 shedding during the outbreak, coupled with proper biosecurity protocols allowed the safe return of the show horses to the event. One of the key elements in reducing the spread of infected horses was monitoring through rectal temperature, early separation and on-site testing of febrile horses. These strategies, while time consuming, rely on the active engagement and communication of all participants. The data show that sporadic febrile horses due to EHV-1 infection are to be expected in a large population of show horses; however, an epidemiological link amongst such horses becomes alarming and indicates direct or indirect transmission.

## Figures and Tables

**Table 1 viruses-17-00608-t001:** Clinical (peak temperature) and EHV-1 qPCR results for nasal secretions from 38 horses involved in an EHV-1 outbreak.

Horse (Age in Years)	Peak Temperature (°C)	qPCR Results for EHV-1 in Nasal Secretions (gB Target Genes/Million Cells)									
		26 January 2025	27 January 2025	28 January 2025	31 January 2025	3 February 2025	11 February 2025	14 February 2025	21 February 2025	27 February 2025	5 March 2025
**1 (6)**	39.0	6023	NA	NA	NA	26,121,278	NA	0	0	0	NA
**2 (8)**	38.5	NA	16,631	NA	NA	114,283	NA	0	0	0	NA
**3 (8)**	38.6	NA	0	NA	22,350	123,270	NA	0	0	0	NA
**4 (8)**	38.6	NA	0	0	7250	53,749	NA	0	0	0	NA
**5 (9)**	38.6	NA	0	NA	0	0	NA	0	0	0	NA
**6 (6)**	38.5	NA	0	NA	0	0	0	NA	0	0	NA
**7 * (6)**	38.5	NA	NA	0	3245	768	NA	0	0	0	NA
**8 (5)**	38.4	NA	NA	0	0	0	NA	1,456,496	0	0	0
**9 (8)**	38.0	NA	NA	NA	NA	0	1888	NA	0	0	NA
**10 (15)**	37.5	NA	NA	NA	NA	0	0	NA	0	0	NA
**11 (8)**	38.0	NA	NA	NA	NA	0	14,959,835	NA	94	0	0
**12 (10)**	38.2	NA	NA	NA	NA	0	NA	40,904	2,606,189	0	0
**13 (8)**	38.2	NA	NA	NA	NA	0	1709	NA	0	0	NA
**14 (18)**	37.9	NA	NA	NA	NA	0	26,636	NA	0	0	NA
**15 (6)**	38.2	NA	NA	NA	NA	27	NA	0	0	0	NA
**16 (10)**	38.3	NA	NA	NA	NA	1.922	NA	0	0	0	NA
**17 (6)**	37.9	NA	NA	NA	NA	948	NA	0	0	0	NA
**18 (13)**	37.9	NA	NA	NA	NA	142,869	NA	0	0	0	NA
**19 * (9)**	38.7	NA	NA	NA	NA	0	998	NA	0	0	NA
**20 (7)**	38.1	NA	NA	NA	NA	0	1,686,166	NA	0	0	NA
**21 (12)**	38.0	NA	NA	NA	NA	0	0	NA	0	0	NA
**22 (9)**	38.0	NA	NA	NA	NA	258,338	NA	0	0	0	NA
**23 (7)**	38.0	NA	NA	NA	NA	0	6720	0	0	0	NA
**24 (7)**	37.9	NA	NA	NA	NA	0	0	NA	0	0	NA
**25 (12)**	37.7	NA	NA	NA	NA	0	0	NA	0	0	NA
**26 (14)**	37.9	NA	NA	NA	NA	0	0	NA	0	0	NA
**27 (5)**	38.0	NA	NA	NA	NA	0	0	NA	0	0	NA
**28 (8)**	38.0	NA	NA	NA	NA	91	NA	0	0	0	NA
**29 (8)**	38.0	NA	NA	NA	NA	0	0	NA	0	0	NA
**30 (8)**	38.0	NA	NA	NA	NA	26	NA	0	0	0	NA
**31 (13)**	38.0	NA	NA	NA	NA	0	0	NA	0	0	NA
**32 (13)**	38.0	NA	NA	NA	NA	44,962	NA	0	0	0	NA
**33 (10)**	37.7	NA	NA	NA	NA	0	NA	0	0	0	NA
**34 (10)**	38.0	NA	NA	NA	NA	0	0	NA	0	0	NA
**35 (8)**	37.8	NA	NA	NA	NA	0	0	NA	0	0	NA
**36 (7)**	37.7	NA	NA	NA	NA	0	0	NA	0	0	NA
**37 (6)**	37.9	NA	NA	NA	NA	0	0	NA	0	0	NA
**38 (10)**	37.9	NA	NA	NA	NA	0	0	NA	0	0	NA

Horse 1 was the first horse with elevated rectal temperature that tested qPCR-positive for EHV-1. Peak rectal temperature represents the highest recorded rectal temperature during the monitoring period. Horses were monitored twice daily. Fever was defined as a rectal temperature ≥ 38.6 °C. NA = not available. * = pony.

## Data Availability

Data available upon request due to privacy restrictions.

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
