# Peer review of "Management of an Equine Herpesvirus-1 Outbreak During a Multi-Week Equestrian Event"

_viruses, 2025, doi:10.3390/v17050608_

Round 1

Reviewer 1 Report

Comments and Suggestions for Authors

The manuscript describes handling with isolation in quarantine of a group of horses during an outbreak of EHV-1 at an equestrian event, to reduce the risk of transmission of EHV-1 to other horses. This is a situation that is of great interest to study, and find useful strategies for, based on evidence and risk assessment. Nine out of 38 horses displayed with clinical signs of fever between January 26 and an unreported date in February, (probably February 14 as mentioned in the Discussion). EHV-1 was detected in nasal swabs in 22 out of 58 samplings on various dates between January 26 and February 21. 
Quarantine was released on February 24 for a number of horses. 
The main issue that needs to be addressed more thoroughly in this report is why it was decided to let horses out from the cohort and back to the show one week after a clinical case, when some horses were still displaying EHV-1 in nasal swabs, pointing to a still ongoing infection in the group. How was it secured that the released horses were not incubating virus? Another relevant question to discuss is the breakthrough of infection in horses presumably vaccinated (vaccine status not described but was regulated by the show arrangers). Also, some statements in the paper seem speculative, such as the effect of antiviral treatments.

L 95-97: please describe the EHV-1 vaccination status of the cohort.

Material and methods- please describe the quarantine and biosecurity routines, e g, separately isolated or as a cohort; indoors/outdoors; were horses allowed body contact, or separated by walls; did they share common air space/separate ventilation; distance between horses; did staff change clothes and shoes between each horse, use disinfected tools for each horse etc. 
Also describe any treatments of horses here. 
Plus, describe how other horses in any adjacent barns at the quarantine facility were protected and monitored.

Material and methods- please describe the method used for genotyping ‘

Material and methods – please describe ethical considerations of the study

L 126: It could help to add the designated horse number (1) to the index case

L129: it could help to add the dates and horse numbers (2-8). 

L 132 “All initial EHV-1 POC PCR results were confirmed via qPCR at a later time and showed 100% agreement.” Please define how many samples that were analysed with both POC PCR and qPCR in parallel,  and the results, and if the numbers allow, Se and Sp for POC PCR. 

L 136-137: Add the date for fever in the additional horse (nr 19) with fever and how many days into quarantine that was, and the result of POC PCR/qPCR.

L 139-141: Please add the dates for the first and last of the 22 PCRpos findings. It would be helpful to include CT values in addition to viral loads estimations. Were both POC PCR and qPCR used at each sample, unclear?

L142: Method for genotyping?

L 144-146: ”During the second week of the outbreak (2.11.25 to 2.14.25) , 9 additional horses tested EHV-1 qPCR-positive; all of 145 them at the exception of one horse displayed normal rectal temperatures” – This is confusing, since the indicated dates Feb 11 and Feb 14 occur during the third week of the outbreak. Also, it would be more informative and honest to describe the main interesting finding clearly: that one more horse showed fever during this time and the date for that.

L 146-148:  Please amend , the date has a typo, and February 21 is during the 4th week

L 148: Please amend, February 27 and March 3 is during the 5th and 6th week of the outbreak. Also, indicate the number of tested horses  (n=16 and n=3 respectively). 

L 149-152: Please explain why only these 17 horses were selected to return to competition and not also other horses with 2 free swabs (number 3,4, 6, 7, 18,21, 23, 25, 26, 27, 32, 34, 35)?
 Please add the information that at this point (Feb 21), there were still 8 horses without 2 negative swabs in the cohort, of which 2 were repeatedly positive for EHV-1 , out of which one had recently showed fever (the date must be added, as previously pointed out).
In the Discussion, please discuss the evidence of still ongoing infection and viral shedding in the cohort at this point of time, and how it was regarded safe to let horses from the cohort leave?

L 160-165: The treatment should suggestedly be moved to Material & methods section since it is not a result. Please indicate what date the treatment started for each treated horse (maybe in table 1?) and comment on the rationale or indication for choosing that date.

L 165: “At the exception of horse 12, all horses treated with 165 valacyclovir experienced a rapid decline in EHV-1 viral load in nasal secretions” – Please show the evidence for this sentence, and discuss the point for treatment in relation to onset of clinical signs or PCR positivity, and what the normal duration of EHV-1 PCR positivity is, and relate the results for the treated horses to the untreated, as the untreated horses that were PCR positive also became PCR negative.
In table 1, the evidence is lacking since the EHV-1 results for:
Horse 1 increases from Jan 26 to Feb 3, and then there is no data until Feb 14. 
Horse 2, increase from Jan 27 to Feb 3, then no data until Feb 14. 
Horses 3 and 4, increase from Jan 31 to Feb 3, then no data until Feb 14.
Horse 8, no data between Feb 14 to Feb 21
Horses 9, 11, 13, 19, 20, 23, no data between Feb 11 to Feb 21
Horse 12, increase from Feb 14 to Feb 21, then no data until Feb 27
Horse 18, 22, 32, no data from Feb 3 to Feb 14
If no evidence can be demonstrated, the text should comment that it remains speculative if the valacyclovir treatment had any effect on EHV-1 shedding.

L 172-178 It is unclear whether the results from monitoring in “adjacent barns” in lines 172-178 relates to barns in the show facilities or in the quarantine facilities. If it would have been in the quarantine, there could be breaches in biosecurity.
Please specify how many horses in the show facilities that were febrile respectively EHV-1 positive after the return of horses from the EHV-1 cohort on February 21, and at what dates. It would also be interesting with information of the location of these cases in relation to the returning horses. How long distance was it between?

Discussion: lacking discussion on this occurrence of EHV-1 spreading in a population of vaccinated horses, and literature references on the efficacy of the vaccine, and vaccine breakthroughs

L 190: Any references in the literature on how often EHV-1 only gives mild or asymptomatic disease?

L 195-198: Speculation? No such data is presented or analyzed, especially the preventive health protocol (EHV vaccination) is lacking. 

L 199: please specify “respiratory viruses”

L 206-208: vaccinated or unvaccinated to EHV-1, any difference in silent shedders?

L 217-219: This sentence is unclear, please clarify what the results of the trailer testing did for the biosecurity? Do you think that the positive result should have led to testing or other increased monitoring of horses from the trailer?

L 220-236: Could serology add anything here, discuss findings in literature

L 249-250: “allowed the horses to resume competition once they all tested negative twice, 7 days apart” -but is seems that “all horses in the cohort had tested negative twice” is not supported by Table 1, see earlier comments.

L 251: here it says Feb 14 for the first time, add to results too?

L 257: please clarify that “The recommended quarantine time of 28 days from resolution of clinical signs from the last infected horse appears to be well justified when dealing with an EHV-1 outbreak situation” was not a recommendation that was followed in this case, since horses were released already Feb 21, only 1 week (?) after the appearance of the last clinical case, and still within the incubation period for EHV-1. How was that justified?

L 270-272: what about serology?

Author Response

The manuscript describes handling with isolation in quarantine of a group of horses during an outbreak of EHV-1 at an equestrian event, to reduce the risk of transmission of EHV-1 to other horses. This is a situation that is of great interest to study, and find useful strategies for, based on evidence and risk assessment. Nine out of 38 horses displayed with clinical signs of fever between January 26 and an unreported date in February, (probably February 14 as mentioned in the Discussion). EHV-1 was detected in nasal swabs in 22 out of 58 samplings on various dates between January 26 and February 21. 
Quarantine was released on February 24 for a number of horses. 
The main issue that needs to be addressed more thoroughly in this report is why it was decided to let horses out from the cohort and back to the show one week after a clinical case, when some horses were still displaying EHV-1 in nasal swabs, pointing to a still ongoing infection in the group. How was it secured that the released horses were not incubating virus? Another relevant question to discuss is the breakthrough of infection in horses presumably vaccinated (vaccine status not described but was regulated by the show arrangers). Also, some statements in the paper seem speculative, such as the effect of antiviral treatments.

Many thanks for brining up this issue. The authors realize that some of the time points were not listed in the results and Table 1 was updated. The timeline may not have been clear; therefore, the authors have highlighting the following timeline:

  • 1/26/25: first case confirmed with EHV-1
  • Quarantine period from 1/28/25 to 2/24/2025
  • Separation of EHV-1 positive horses in nasal secretions
  • Two negative nasal swabs, taken 5-7 days apart (one shedding period) were requested to determine a horse free of EHV-1 shedding
  • All horses tested negative on 2/27/25
  • Horse 12 was separated and placed on antiviral drugs
  • Return of 17 horses to the show (only horses with 2 negative nasal swabs, 7 days apart)

The authors reviewed the vaccination dates from the 38 horses. All horses had been vaccinated with a killed-adjuvanted EHV-1/EIV vaccine 17 to 155 days prior to the onset of the outbreak (median of 69 days). The missing information has been added in the manuscript.

The speculative statement regarding the use of antiviral drugs were removed.

L 95-97: please describe the EHV-1 vaccination status of the cohort.

The missing information pertaining to the vaccination history has been added, i.e., “The vaccination history of the 38 horses consisted in the administration of killed-adjuvanted multivalent EHV-1/EIV vaccine (brand unknown) 17-155 days (median 69 days) prior to arrival at the show.

Material and methods- please describe the quarantine and biosecurity routines, e g, separately isolated or as a cohort; indoors/outdoors; were horses allowed body contact, or separated by walls; did they share common air space/separate ventilation; distance between horses; did staff change clothes and shoes between each horse, use disinfected tools for each horse etc. 

The missing information has been added as requested by the reviewer, i.e., “The facility was composed of two multi-stall barns. Horses were stabled indoors based on their EHV-1 shedding status with EHV-1 qPCR-positive horses in one barn and EHV-1 qPCR-negative horses in the other barn. In each barn, horses were separated by solid walls but shared common airspace. Proper biosecurity protocols, including foot bath at the entrance of the barns, dedicated personal protective equipment (booties, coveralls, disposable gloves) and separated stall cleaning equipment were used to reduce indirect transmission amongst the study horses.”
Also describe any treatments of horses here.

The information pertaining to the treatment has been added under material and methods.

Plus, describe how other horses in any adjacent barns at the quarantine facility were protected and monitored.

The quarantine facility was an empty facility that only housed the cohort of 38 horses involved in the outbreak. However, at the show event, all competing show horses were monitored daily for elevated rectal temperature and such horses were rapidly separated and tested for EHV-1 by qPCR.

Material and methods- please describe the method used for genotyping ‘

The genotype was determined by targeting the ORF 30 gene and the single nucleotide polymorphism at position 2254 (A versus G) of the nucleotide sequence. A reference of the assay has been added under material and methods.

Material and methods – please describe ethical considerations of the study

As this study used biological samples collected for diagnostic reasons without any direct actions from the authors’ side influencing the management of each study horse, institutional animal care and use committee evaluation was not required. Owner and trainer consent for the collection of the nasal secretions was requested.

L 126: It could help to add the designated horse number (1) to the index case

The horse number has been added.

L129: it could help to add the dates and horse numbers (2-8). 

The horse numbers have been added.

L 132 “All initial EHV-1 POC PCR results were confirmed via qPCR at a later time and showed 100% agreement.” Please define how many samples that were analysed with both POC PCR and qPCR in parallel,  and the results, and if the numbers allow, Se and Sp for POC PCR. 

Only the initial samples from the 8 horses were tested in parallel using qPCR and the POC PCR assay.

L 136-137: Add the date for fever in the additional horse (nr 19) with fever and how many days into quarantine that was, and the result of POC PCR/qPCR.

The date has been added. As horses were moved on 2/28/25 to the empty facility, this would have been 6 days into quarantine. Only the initial 8 samples were run in parallel with the POC PCR assay. The subsequent samples were only run using qPCR.

L 139-141: Please add the dates for the first and last of the 22 PCRpos findings. It would be helpful to include CT values in addition to viral loads estimations. Were both POC PCR and qPCR used at each sample, unclear?

See statement above regarding the use of qPCR testing for all samples once horses were into quarantine. The date range of qPCR positive horses has been added, i.e., 1/26/2025 to 2/14/2025. The use of Ct or Cq values should not be used in the opinion of the authors as it is biased by sample quality, nucleic acid purification and performance of the assay. Values that are normalized against a housekeeping gene have repeatedly been shown to be more accurate and allow comparison amongst results.

L142: Method for genotyping?

The methods, i.e. reference to assay targeting the ORF 30 of EHV-1, was added under material and methods (line 128).

L 144-146: ”During the second week of the outbreak (2.11.25 to 2.14.25) , 9 additional horses tested EHV-1 qPCR-positive; all of 145 them at the exception of one horse displayed normal rectal temperatures” – This is confusing, since the indicated dates Feb 11 and Feb 14 occur during the third week of the outbreak. Also, it would be more informative and honest to describe the main interesting finding clearly: that one more horse showed fever during this time and the date for that.

The horse (19) with elevated rectal temperature was added to prevent any confusion.

L 146-148:  Please amend , the date has a typo, and February 21 is during the 4th week

The sentence was amended as requested by the reviewer.

L 148: Please amend, February 27 and March 3 is during the 5th and 6th week of the outbreak. Also, indicate the number of tested horses  (n=16 and n=3 respectively). 

The corrections were addressed as suggested.

L 149-152: Please explain why only these 17 horses were selected to return to competition and not also other horses with 2 free swabs (number 3,4, 6, 7, 18,21, 23, 25, 26, 27, 32, 34, 35)?

The decision to bring back horses and complete the show circuit was left at the trainer’s discretion.

 Please add the information that at this point (Feb 21), there were still 8 horses without 2 negative swabs in the cohort, of which 2 were repeatedly positive for EHV-1 , out of which one had recently showed fever (the date must be added, as previously pointed out).
On the date of return to competition (2/24/25)  all horses tested qPCR-negative in nasal secretions with 3 horses having only 1 negative nasal swab taken. These three horses were tested one week later (3/5/25) and while negative, never returned to the show.

In the Discussion, please discuss the evidence of still ongoing infection and viral shedding in the cohort at this point of time, and how it was regarded safe to let horses from the cohort leave?

In general, and with appropriate biosecurity protocols, regulatory officials required a quarantine period of 28 days. The period of 28 days is also recommended in the updated EHV-1 consensus statement. With the introduction of qPCR and repeated testing, the release of horses from quarantine/isolation can be performed following two EHV-1 qPCR-negative nasal swabs taken 5-7 days apart. Because all EHV-1 qPCR-positive horses were moved into a different barn and separated from the horses that never became EHV-1 qPCR-positive, and the repeated negative testing of the latter cohort of horses, it is safe to assume that no virus was circulating in these horses.

L 160-165: The treatment should suggested to be moved to Material & methods section since it is not a result. Please indicate what date the treatment started for each treated horse (maybe in table 1?) and comment on the rationale or indication for choosing that date.

The information pertaining to the treatment was moved into material and methods. The rational to treat horses at onset of the first EHV-1 qPCR-positive results was left to the discretion of the owners and attending veterinarians. Previous observations have shown that the treatment of horses with no evidence of viremia or nasal shedding only prevents infection during the treatment period. The overall consensus on preventive treatment applies to horses during the early stage of the disease (fever, viremia, viral shedding). Additional information has been added to prevent any confusion and acknowledging that the decision to treat was arbitrarily chosen to minimize potential complications (EHM).

L 165: “At the exception of horse 12, all horses treated with valacyclovir experienced a rapid decline in EHV-1 viral load in nasal secretions” – Please show the evidence for this sentence, and discuss the point for treatment in relation to onset of clinical signs or PCR positivity, and what the normal duration of EHV-1 PCR positivity is, and relate the results for the treated horses to the untreated, as the untreated horses that were PCR positive also became PCR negative.

In table 1, the evidence is lacking since the EHV-1 results for:
Horse 1 increases from Jan 26 to Feb 3, and then there is no data until Feb 14. 
Horse 2, increase from Jan 27 to Feb 3, then no data until Feb 14. 
Horses 3 and 4, increase from Jan 31 to Feb 3, then no data until Feb 14.
Horse 8, no data between Feb 14 to Feb 21
Horses 9, 11, 13, 19, 20, 23, no data between Feb 11 to Feb 21
Horse 12, increase from Feb 14 to Feb 21, then no data until Feb 27
Horse 18, 22, 32, no data from Feb 3 to Feb 14
If no evidence can be demonstrated, the text should comment that it remains speculative if the valacyclovir treatment had any effect on EHV-1 shedding.

Horse 12 was unusual as the viral load continued to increase while being treated with valacyclovir. This so perceived treatment failure was further investigated and thought to relate to the inability to administer the entire amount of valacyclovir to this horse. Little information is available on the use of valacyclovir and its efficiency at reducing levels of EHV-1 viral shedding and viremia. Previous work has shown that surviving field cases infected with EHV-1 and treated with valacyclovir show a rapid decline in viral shedding within 5 days of treatment (Estell KE, Dawson DR, Magdesian KG, Swain E, Laing ST, Siso S, Mapes S, Pusterla N. Quantitative molecular viral loads in 7 horses with naturally occurring equine herpesvirus-1 infection. Equine Vet J. 2015 Nov;47(6):689-93). In the present study and since the intervals of EHV-1 testing was an average of 7 days apart, the efficiency of valacyclovir in reducing viral shedding can not be proven. The statement in the discussion about the use of valacyclovir has been changed to highlight the lack of benefit of using valacyclovir in reducing environmental contamination.

L 172-178 It is unclear whether the results from monitoring in “adjacent barns” in lines 172-178 relates to barns in the show facilities or in the quarantine facilities. If it would have been in the quarantine, there could be breaches in biosecurity.

The monitoring was performed at the show event and targeted febrile horses prior, during and after the EHV-1 outbreak.
Please specify how many horses in the show facilities that were febrile respectively EHV-1 positive after the return of horses from the EHV-1 cohort on February 24, and at what dates. It would also be interesting with information of the location of these cases in relation to the returning horses. How long distance was it between?

The rate of EHV-1 qPCR-positive febrile horses after February 24th (during the last 2 weeks of the circuit) was 1/3 febrile horses. None of the additional EHV-1 qPCR-positive horses appear to have any epidemiological link to the outbreak horses. 

Discussion: lacking discussion on this occurrence of EHV-1 spreading in a population of vaccinated horses, and literature references on the efficacy of the vaccine, and vaccine breakthroughs

The authors thank the reviewer for bringing up the sticky point of vaccination. The recent review article by Osterrieder et al. (Vaccination for the prevention of equine herpesvirus-1 disease in domesticated horses: A systematic review and meta-analysis. J Vet Intern Med. 2024 May-Jun;38(3):1858-1871) indicates that commercial and experimental vaccines minimally reduce the incidence of clinical disease associated with EHV-1 infection. The EHV-1 vaccines used in the US are considered aid in disease prevention with the expected benefit of reducing clinical disease and also reduce viral shedding. A sentence has been added in the discussion to highlight this point.

L 190: Any references in the literature on how often EHV-1 only gives mild or symptomatic disease?

A reference has been added.

L 195-198: Speculation? No such data is presented or analyzed, especially the preventive health protocol (EHV vaccination) is lacking. 

The statement was changed to a speculation in order to soften the sentence.

L 199: please specify “respiratory viruses”

The specific respiratory viruses have been added

L 206-208: vaccinated or unvaccinated to EHV-1, any difference in silent shedders?

All studies available on this topic involve show horses, which are required to be vaccinated against EHV-1 in the US.

L 217-219: This sentence is unclear, please clarify what the results of the trailer testing did for the biosecurity? Do you think that the positive result should have led to testing or other increased monitoring of horses from the trailer?

The positive trailer testing did not change the biosecurity protocols and requirement to monitor horses. It only increased the awareness of the show organizer on this specific horse population (i.e. the one from the positive trailer).

L 220-236: Could serology add anything here, discuss findings in literature

Very good point that the reviewer is bringing up. In hindsight the collection of serum samples at the onset of the first febrile horse could have helped determine the risk of EHV-1 infection based on total anti-EHV-1 IgG and IgG 4/7. Recent work Eady et al. (Equine herpesvirus type 1 (EHV-1) replication at the upper respiratory entry site is inhibited by neutralizing EHV-1-specific IgG1 and IgG4/7 mucosal antibodies. J Virol 2024, 98, e0025024) has shown that robust mucosal immunity can be essential in protecting against EHV-1 and to reduce EHM outbreaks. The lack of blood/mucosal anti-EHV-1 antibody detection was added in the limitation section of the discussion.

L 249-250: “allowed the horses to resume competition once they all tested negative twice, 7 days apart” -but is seems that “all horses in the cohort had tested negative twice” is not supported by Table 1, see earlier comments.

The return to competition was only allowed to horses that tested EHV-1 qPCR negative twice 7 days apart. The sentence states that horses were allowed to resume competition once they tested negative twice, 7 days apart.

L 251: here it says Feb 14 for the first time, add to results too?

This is presented under results as 9 additional horses tested qPCR positive in the 2nd and 3rd week. Horse 8 and 12 tested qPCR-positive for EHV-1 for the first time on 2/14/2025.

L 257: please clarify that “The recommended quarantine time of 28 days from resolution of clinical signs from the last infected horse appears to be well justified when dealing with an EHV-1 outbreak situation” was not a recommendation that was followed in this case, since horses were released already Feb 21, only 1 week (?) after the appearance of the last clinical case, and still within the incubation period for EHV-1. How was that justified?

The recommendation of 28 days applies to outbreaks that are not monitored via qPCR and according to the ACVIM consensus statement (reference), a quarantine of up to 28 days is justified. However, with the introduction of qPCR testing, and the isolation of shedders, the quarantine can be shortened as long as non-clinical horses with no further exposure teste qPCR-negative for EHV-1 twice, 5-7 days apart. The decision to lift a quarantine depends on the reportability of the disease and remains the decision of the regulatory officials.

L 270-272: what about serology?

We have added this limitation in the paragraph.

Reviewer 2 Report

Comments and Suggestions for Authors

The authors are to be complimented on their study and clear an concise presentation of their data.

Lines 73, a9a: When discussing temperature monitoring you mention daily measurements: both USEF (www.usef.org/forms-pubs/RDXPCI76Fg8/temperature-log) and FEI (https://inside.fei.org/fei/ehv-1/biosecurity-how-to/temperature) recommend twice daily measurement with FEI also requiring twice daily measurement of temperature for the 3 days prior to arrival at the site of an equestrian event. In addition, USEF included temperature-sensing microchips in their guidance when microchipping was introduced by them in 2023  (see www.usef.org/media/press-releases/us-equestrian-introduces-microchip-rule-for). USEF requires rectal temperature exceeding 38.6C/101.5F to be reported to a veterinarian and/or competition management. How often was rectal temperature measured in the horses in the study?

Table 1. This is very interesting indeed: if rectal temperature was only measured once daily is it possible that this was underestimating peak rectal temperature, for all horses with a positive tier except #1 and #19 (and perhaps #3 and #4), which would exceed the threshold proposed by USEF.

Table 1: Consider putting the results in order of rectal temperature and/or of highest qPCR result rather than by horse number: that might show the pattern of response perhaps in relation to peak rectal temperature but also time of highest titer. Consider adding the age of the horse as a variable given the discussion of rectal temperature in relation to age.

Line 225: Please discuss the role of immunity in relation to rectal temperature and the potential negative impact on antiviral treatment on the development of immune response following infection. Please also discuss the role of vaccination history (with age) in relation to this. Does your data suggest that the duration of shedding was decreased by antiviral treatment, given that it is both amount of virus shed and duration that play a role in risk of transmission and spread of infection

Line 234 please be more specific around appropriate biosecurity and frequency of taking temperature measurements

Please update the literature search to include recent articles on biosecurity in equines: for example

Cross-Sectional Survey of Horse Owners to Assess Their Knowledge and Use of Biosecurity Practices for Equine Infectious Diseases in the United States - PubMed

Long-term performance of show-jumping horses and relationship with severity of ataxia and complications associated with myeloencephalopathy caused by equine herpes virus-1 - PubMed

Pharmacologic interventions for the treatment of equine herpesvirus-1 in domesticated horses: A systematic review - PubMed

Vaccination for the prevention of equine herpesvirus-1 disease in domesticated horses: A systematic review and meta-analysis - PubMed

Author Response

Lines 73, a9a: When discussing temperature monitoring you mention daily measurements: both USEF (www.usef.org/forms-pubs/RDXPCI76Fg8/temperature-log) and FEI (https://inside.fei.org/fei/ehv-1/biosecurity-how-to/temperature) recommend twice daily measurement with FEI also requiring twice daily measurement of temperature for the 3 days prior to arrival at the site of an equestrian event. In addition, USEF included temperature-sensing microchips in their guidance when microchipping was introduced by them in 2023  (see www.usef.org/media/press-releases/us-equestrian-introduces-microchip-rule-for). USEF requires rectal temperature exceeding 38.6C/101.5F to be reported to a veterinarian and/or competition management. How often was rectal temperature measured in the horses in the study?

Many thanks for bringing up the USEF and FEI regulations. While USEF cannot reinforce the recommendations, the trainers/owner present at the show have been sensitized by the 2021/2022 EHM outbreak and have learned their lesson. The compliance for twice daily rectal temperature assessment is high at this show circuit.

Table 1. This is very interesting indeed: if rectal temperature was only measured once daily is it possible that this was underestimating peak rectal temperature, for all horses with a positive tier except #1 and #19 (and perhaps #3 and #4), which would exceed the threshold proposed by USEF.

The rectal temperatures were taken twice daily.

Table 1: Consider putting the results in order of rectal temperature and/or of highest qPCR result rather than by horse number: that might show the pattern of response perhaps in relation to peak rectal temperature but also time of highest titer. Consider adding the age of the horse as a variable given the discussion of rectal temperature in relation to age.

The age of the horses has been added. Unfortunately, the reordering of the horses was not possible as it conflicted with requirements from reviewer 1.

Line 225: Please discuss the role of immunity in relation to rectal temperature and the potential negative impact on antiviral treatment on the development of immune response following infection. Please also discuss the role of vaccination history (with age) in relation to this. Does your data suggest that the duration of shedding was decreased by antiviral treatment, given that it is both amount of virus shed and duration that play a role in risk of transmission and spread of infection.

A sentence and reference were added regarding immunity, specifically the use of commercial vaccines, and the positive effect on disease expression. There is no to little information pertaining to the impact of anti-herpetic drugs and their impact on the immune response. While antiviral drugs can rapidly reduce viral loads in blood and nasal secretions, they are generally given at onset of early signs, i.e. a times when viremia and nasal shedding are already present and have triggered an anamnestic immune response. Unfortunately, because this is a case repot with no untreated control horses expressing the same levels of viral shedding as treated horses, the effect of antivirals on viral kinetics cannot be assessed. A statement was added in the discussion regarding this point.

Line 234 please be more specific around appropriate biosecurity and frequency of taking temperature measurements

Information pertaining to proper biosecurity measures were added under material and methods as  well as the importance of monitoring rectal temperature twice daily.

Please update the literature search to include recent articles on biosecurity in equines: for example

Additional references were added.